# Efficacy of Spinetoram for the Control of Bean Weevil, *Acanthoscelides obtectus* (Say.) (Coleoptera: Chrysomelidae) on Different Surfaces

**DOI:** 10.3390/insects13080723

**Published:** 2022-08-12

**Authors:** Özgür Sağlam, Ahmet Çelik, Ali Arda Işıkber, Hüseyin Bozkurt, Maria K. Sakka, Christos G. Athanassiou

**Affiliations:** 1Plant Protection Department, Agriculture Faculty, Tekirdağ Namık Kemal University, Değirmenaltı Campus, Tekirdağ 59030, Turkey; 2Plant Protection Department, Agriculture Faculty, Kahramanmaraş Sütçü İmam University, Avşar Campus, Kahramanmaraş 46100, Turkey; 3Laboratory of Entomology and Agricultural Zoology, Department of Agriculture, Crop Production and Rural Environment, University of Thessaly, 38446 Nea Ionia, Greece

**Keywords:** spinetoram, *Acanthocelides obtectus*, mortality, surface treatment, contact toxicity

## Abstract

**Simple Summary:**

Contact toxicity of spinetoram on three different surfaces, concrete, ceramic floor tile and laminate flooring, against *Acanthocelides obtectus* (Say.) was evaluated in laboratory bioassays. Our results provide data on the insecticidal effect of spinetoram for the control of *A. obtectus* on various surfaces; however, its efficacy varies according to the surface type, exposure time and concentration. In conclusion, our laboratory tests indicated that spinetoram at 0.025 and 0.05 mg active ingredient (AI)/cm^2^ achieved satisfactory control at relatively short exposures by contact action of *A. obtectus* adults on three surfaces, commonly encountered in legume storage facilities and warehouses.

**Abstract:**

In this study, the contact toxicity of spinetoram on three different surfaces, concrete, ceramic floor tile and laminate flooring, against *Acanthocelides obtectus* (Say.) (Coleoptera: Chrysomelidae) was evaluated in laboratory bioassays. Different concentrations were evaluated ranging from 0.0025 to 0.05 mg AI/cm^2^, against adults of *A. obtectus*. Adult mortality was measured after 1-, 3-, 5- and 7-day exposure. After 1-day exposure, the mortality was low on all surfaces, ranging from 0 to 27.2%. After 5- and 7-day exposure, spinetoram at concentrations of 0.01 mg/cm^2^ and above achieved 100% or close mortality on concrete and laminate flooring surface, whereas low concentrations (0.0025, 0.005 and 0.0075 mg AI/cm^2^) resulted in significantly lower mortality levels, ranging from 1.6 to 30.8%, than high concentrations. In the case of ceramic floor tile surface, spinetoram treatments at all tested concentrations did not result in 100% mortality. Significant differences were recorded among the surfaces, depending on concentrations and exposure intervals. After 3-, 5- and 7-day exposure, mortality levels on ceramic floor tile surface were generally higher at low concentrations than those on the concrete and laminate flooring surfaces, whereas those on concrete and laminate flooring surfaces were significantly higher at high concentrations than ceramic floor tile surface. These results indicate that spinetoram at 0.025 and 0.05 mg AI/cm^2^ achieve satisfactory control at relatively short exposures on common types of surfaces and thus can be used as an effective insecticide against *A. obtectus.*

## 1. Introduction

Common bean (*Phaseolus vulgaris* L.) is a major grain legume consumed worldwide for its edible seeds and pods, since it is an important source of protein, vitamins, carbohydrates, etc., and provides 15% of protein and fulfills 30% of caloric requirements of the global population [1,2]. The worldwide production of common bean was 34.8 million ha, and production was 27.5 million tons in 2020 [3]. Dry bean production exists worldwide and effected by various biotic and abiotic parameters. Postharvest losses are major biotic parameters caused by the bruchid species *Acanthoscelides obtectus* (Say) (Coleoptera: Chrysomelidae) and *Zabrotes subfasciatus* (Boheman) (Coleoptera: Chrysomelidae). The common bean weevil, *A. obtectus,* is a devastating insect pest, capable of causing severe bean crop losses in different areas, such as America [4,5], Australia [6], Africa [7,8], the Mediterranean area, [9,10] and Europe [11,12].

Populations of *A. obtectus* are mostly found in stores of dried legumes and can adapt well for reproduction in a storage environment. The first instar larvae burrows into the seed to feed and metamorphose from larva to adult [13,14]. Larvae are able to destroy embryos and directly affect germination, thus causing losses in stored beans [15,16,17,18,19]. Seed losses can range from 7–40%, which means 1.59–9.12 million tons of damaged seeds each year in the world, caused by bruchids [20]. This equates to a loss of 1.59–9.12 million tons each year in the world, caused by bruchids. Chemical control is the most common practice of managing bean weevils using residual synthetic insecticides, such as organophosphates and pyrethroids, and fumigants, such as phosphine [21,22,23,24,25]. However, the use of these insecticides has some drawbacks, e.g., negative impact on products, high toxicity to humans and animals, the presence of residues and/or resistance of pests to pesticides [26,27,28,29]. Besides, the use of organophosphate insecticides, such as malathion, chlorpyrifos-methyl and pirimiphos-methyl in stored products are facing restriction and may be eliminated from the market [28,30]. Pyrethroids can be used as an alternative to some of the traditional organophosphates, due to their low odor, quick action and low human toxicity [26,31]. However, the repeated use of pyrethroid-based insecticides has resulted in resistance issues in many parts of the world [32,33,34,35,36]. An alternative pest management strategy or new compounds that are eco-friendly and in accordance with the food safety requirements are needed to control *A. obtectus* in the common bean.

One of the most promising alternatives on stored products is spinosad, which is a metabolite of the actimomycete *Saccharopolyspora spinosa* Mertz and Yao (Bacteria: Actinobacteridae) and consists of a mixture of spinosyn A and spinosyn D [37,38]. It has low mammalian toxicity [38,39,40] and has been reported to provide effective and long-lasting control of numerous key stored product insects on various commodities [41,42,43,44,45,46,47,48,49,50,51]. Spinosad was registered for use in stored products in the USA and a number of other countries as an alternative to traditional grain protectants. Spinetoram, a semisynthetic active ingredient in the spinosyn family (a mixture of spinosyn J and spinosyn L), was introduced as an insecticide with greater potency and faster speed of action on various field insect pests in comparison with spinosad [52,53]. Spinetoram was found effective against a variety of stored grain beetle species, e.g., wheat, oat and paddy rice, and stable to different temperatures and relative humidity levels under laboratory conditions [54,55,56]. Spinetoram has been tested against different stored-product species and found more effective than spinosad [54,57].

As a surface insecticide, spinetoram was also found to be very effective against adults and young larvae of the confused flour beetle, *Tribolium confusum* Jacquelin du Val (Coleoptera: Tenebrionidae) [58,59]. Vassilakos and Athanassiou [55] reported that spinetoram was also effective against the adults of *Rhyzopertha dominica* (F.) (Coleoptera: Bostrychidae), *Sitophilus oryzae* (L.) (Coleoptera: Curculionidae), *Sitophilus granarius* (L.) (Coleoptera: Curculionidae), *T. confusum*, *Oryzaephilus surinamensis* (L.) (Coleoptera: Sivanidae), and *Cryptolestes ferrugineus* (Stephens) (Coleoptera: Laemophloeidae) on several types of surfaces (concrete, ceramic tile, galvanized steel and plywood), and the presence of food did not influence its efficacy in most cases.

In surface treatments, efficacy and persistence of insecticides is influenced by various factors, such as the type of the surface, tested active ingredients and insect species, temperatures, and relative humidity and light intensity (e.g., sun light, UV light and indoor light) [60,61,62,63]. In the study of Vassilakos and Athanassiou [55], among the surfaces, differences in spinetoram toxicity were noted only for *T. confusum* out of six stored-grain insect species and adult mortality on concrete and galvanized steel was higher than on ceramic tile and plywood in the absence of food. However, to our knowledge, the efficacy of spinetoram as a contact insecticide in surfaces against *A. obtectus* has not been tested so far. In this study, we tested spinetoram in surface treatments against adults of *A. obtectus* under laboratory conditions.

## 2. Materials and Methods

### 2.1. Test Insect

The original population of *A. obtectus* was collected from dry common beans processing companies in Mersin province (Turkey) during the year 2018. The common bean (*P. vulgaris*) with the local cultivar of “Şehirali-90” (11.8% moisture content) was used to feed the *A. obtectus* adults whose were put into glass jars (150 mm in diameter and 250 mm high) and covered with a cloth to allow aeration. The initial population started with at least 300 individuals per kg of bean grains and was reared in a growth chamber under a 12: 12 h light: dark photoperiod, at 27 ± 1 °C and 65 ± 5% r.h. In order to avoid possible insect infestations from the field, the bean grains were kept at ambient conditions of −10 °C for 14 days prior to be used for the culture of *A. obtectus*. One–two days old adults obtained from stock insect culture and pesticide/insect-free bean seeds were used in the biological tests.

### 2.2. Tested Insecticide

A formulation of spinetoram (Delegate 250 WG) with 250 g of active ingredient (AI, spinetoram) per liter was supplied by Dow Agro Sciences. Spinetoram was diluted in distilled water for various surface treatments. Delegate 250 WG has been registered against several insect pests in pear, apple, grape, cotton, maize, pistachio and cherry in many countries, as well as Turkey and Greece.

### 2.3. Surfaces

In biological tests, three different surfaces (concrete, ceramic floor tile and laminate flooring) were used. For preparation of concrete surface; the mortar consisted of mixture of 200 g cement and 40 mL water was prepared and poured into the plastic boxes (100 × 100 × 60 mm^3^) with a thickness of 2 cm. Thereafter, the mortar in the plastic boxes was allowed to dry for 48 h at room temperature, and afterwards the surfaces were heated at 100 °C to cure in the drying oven (Memmert UN30, Memmert GmbH & Co., Schwabach, Germany) for 72 h before use in trials. The ceramic floor tiles (TOPAZ, Yurtbay Ceramic Trade Inc., Eskişehir, Turkey) were purchased at a local hardware store and cleaned with an industrial floor cleaner (KLORAK FT, Klorak Chemicals and Cleaning Products Industry Trade Inc.İzmir, Turkey), according to the manufacturer’s instructions. The tiles used in the biological tests are made of a mixture of clay, kaolin, quartz, feldspat and limestone. The tempering is carried out at a temperature above 900 °C and reheated at 1100 °C during processing of the tiles with 150 × 150 × 5.5 mm^3^ size, according to Turkish Standards (TS 202). The tiles were cut to size of 100 × 100 mm^2^ used in the trials by tile cutting tool and placed into plastic boxes. The laminate flooring used in biological tests is in accordance with EN 717 E-1 standards, with increased resistance to moisture (HDF), and in dimensions of 8 × 195 × 1200 mm^3^. The dimensions of 100 × 100 mm^2^ laminate flooring used in trials were cut with laminate flooring cutter and placed in plastic boxes. The gaps in the margin of ceramic floor and laminate flooring in plastic boxes were filled with hot silicone glue (Dremel^®^ high Temperature Glue Stick, Dremel Company, Racine, WI, USA) in the thin stripe layer.

### 2.4. Treatment of Surfaces and Insect Exposure

Biological tests were carried out on a concrete, ceramic floor tile and laminated flooring without the commodity at 26 ± 1 °C temperature and 65 ± 5% r.h. in a completely dark condition. Each replication of the three surface types was treated with 1 mL of distilled water (control treatment) or an aqueous suspension (1 mL) of spinetoram to provide deposits of 0.0025, 0.005, 0.0075, 0.01, 0.015, 0.025 and 0.05 mg AI/cm^2^. Surfaces were sprayed with distilled water and spinetoram suspension by using an artist’s airbrush HSENG Airbrush AS18 model, (Ningbo Haosheng Pnömatik Machinery Co., Ningbo, China) connected to an air compressor providing 20-psi pressure. Separate airbrush guns were used for distilled water and spinetoram treatments. After treatment, surfaces were air dried for 12 h at room temperature in a laminar flow hood. Twenty-five adults of *A. obtectus,* one- or two-day-old, were introduced into each separate water- and spinetoram-treated surface. In the biological tests, concrete, ceramic floor tile and laminated flooring surface in plastic boxes with surface area of 100 cm^2^ were used for each replicate. Each trial was replicated 5 times and 5 controls were left for each treatment. Surfaces were allowed to dry for 24 h at 25 ± 1 °C temperature and 50 ± 10% r.h. A thin coating of Fluon^®^ (Polytetra-fluoroethylene, Sigma Aldrich product number 665800, Dorset, UK) was applied to the interior side-wall of each arena using a small paintbrush in order to prevent the insects escaping from the experimental arenas.

Mortality was assessed after 1, 3, 5 and 7 days of exposure. Alive (mobile) and death adults on surfaces were observed under a stereoscope in the treated and untreated surfaces. With the term alive we characterized the adults that were able to walk normally. The adults that were not mobile and had no visible movements in their legs and antennae were characterized as death. At each count, dead adults on the surface were removed from the experimental area with a small brush, and alive ones were left on spinetoram treated-surface for further counts.

### 2.5. Experimental Design and Data Analysis

The experiment was designed as factorial, based on completely randomized design with five replications. The experimental factors comprised surfaces at three levels and spinetoram concentration at seven levels and distilled water treatment on the surfaces was considered as a control. The response variables were percentage that was dead after exposure period of spinetoram treatment. All percentages (x) were corrected for mortality on control surfaces [64] and transformed to arcsine (x)^0.5^ [65] to normalize heteroscedastic treatment variances. Mortality data were analyzed by using two-way ANOVA, while means were separated by Tukey test at 5% significant level using general linear models (GLM) of SAS [66].

## 3. Results

### 3.1. Efficacy on Concrete Surface

The two-way ANOVA analysis indicated that all main effects (surface and concentration) and their associated interactions for each exposure interval were significant (Table 1).

Mortality on concrete surface was significantly affected by spinetoram concentration for each exposure interval (Table 2).

After 1-day exposure, mortality was very low, ranging from 0 to 27% and there were no significant differences in mortality levels among the concentrations, except concentration of 0.025 and 0.05 mg AI/cm^2^. After 3-day exposure, mortality level increased significantly with increasing concentration and the high concentrations (0.025 and 0.05 mg AI/cm^2^) resulted in significantly higher mortality with 91.2 and 92%, respectively, than the low concentrations (0.0025, 0.005, 0.0075 and 0.01 mg AI/cm^2^). However, none of the concentrations achieved the complete mortality (100%). After 5- and 7-day exposure, spinetoram treatments at 0.01 mg AI/cm^2^ and above concentrations resulted in 100% or close mortality (Table 2), whereas the low concentrations (0.0025, 0.005 and 0.0075 mg AI/cm^2^) resulted in significantly lower mortality rates, ranging from 8 to 30.8%, than the high concentrations.

### 3.2. Efficacy on Laminate Flooring Surface

Mortality of *A. obtectus* adults for laminate flooring surface was also significantly affected by spinetoram concentration for each exposure interval (Table 3).

After 1-day exposure, mortality was very low, ranging from 0 to 24.8%; no significant differences were noted in mortality levels at the concentrations ranging from 0.0025 to 0.01 mg AI/cm^2^. After 3-day exposure, the high concentrations (0.015, 0.025, and 0.05 mg AI/cm^2^) resulted in significantly higher mortality with 80.8 and 84.8%, respectively, than the low concentrations (0.0025, 0.005, 0.0075 and 0.01 mg AI/cm^2^). However, none of the concentrations achieved the complete mortality. After 5- and 7-day exposure, spinetoram treatments at 0.015 mg AIcm^2^ and above concentrations resulted in 100% or close mortality (Table 3), whereas the low concentrations (0.0025, 0.005 and 0.0075 mg AI/cm^2^) resulted in significantly lower mortality rates ranging from 1.6 to 13.6% than the high concentrations.

### 3.3. Efficacy on Laminate Flooring Surface

Mortality of *A. obtectus* adults for ceramic tile surface was also significantly affected by spinetoram concentration for each exposure interval (Table 4).

After 1-day exposure, the mortality of *A. obtectus* was extremely low, ranging from 0 to 16.1% while there were no significant differences in mortality rates at the concentrations ranging from 0.0025 to 0.015 mg AI/cm^2^. After 3-day exposure, none of the concentrations achieved the complete mortality and the highest mortality with 53.1% was achieved at 0.025 mg AI/cm^2^, which was not significantly different than those at 0.015 and 0.05 mg AI/cm^2^. After 5- and 7-day exposure, spinetoram treatments at all tested concentrations did not result in 100% mortality (Table 4). Whereas after 7-day of exposure, the high concentrations with 0.015 mg AI/cm^2^ ≤ resulted in nearly 100% mortality (≤92%), which were significantly higher than at the low concentrations (0.0025 and 0.005 mg AI/cm^2^).

### 3.4. Comparison of Tested Surfaces

The mortality of *A. obtectus* adults significantly varied by surface type for each exposure interval depending on spinetoram concentrations (Figure 1). The mortality of *A. obtectus* after 1 day of exposure on all treated surfaces was very low, ranging from 0 to 27.2%. There were no differences in mortality levels among the surfaces at 0.0025, 0.005, 0.0015 and 0.025 mg AI/cm^2^, while the mortality levels at 0.0075, 0.01 and 0.05 mg AI/cm^2^ were significantly higher on the treated concrete surface than on the ceramic floor tile surface. After 3-day exposure, at the low concentrations (0.0025, 0.005 and 0.0075 mg AI/cm^2^), mortality levels on the ceramic floor tile surface were generally higher than those on the concrete and laminate flooring surfaces, whereas, at the high concentrations (0.015, 0.025 and 0.05 mg AI/cm^2^), those on the concrete and laminate flooring surfaces were significantly higher than ceramic floor tile surface. Although spinetoram treatment on the concrete surface generally achieved higher mortality levels than the laminate flooring surface, with the exception of spinetoram concentration of 0.015 mg AI/cm^2^, there were not significant differences in mortality levels at between the concentrations of 0.0025, 0.005, 0.015, 0.05 mg AI/cm^2^. The highest mortality (92%) was noted at the concrete surface at 0.025 mg/cm^2^ after 3 days of exposure. Similarly, after 5-day exposure, the low concentrations (0.0025, 0.005 and 0.0075 mg AI/cm^2^) resulted in significantly higher mortality levels on the ceramic floor tile surface than those on the concrete and laminate flooring surfaces. Whereas, at the high concentrations (0.015, 0.025 and 0.05 mg AI/cm^2^) the concrete and laminate flooring surfaces had significantly higher mortality levels than the ceramic floor tile surface. On the other hand, there were no significant differences in mortality levels between the concrete and laminate flooring surface. The complete mortality was achieved only on the concrete and laminate flooring surface at 0.025 and 0.05 mg AI/cm^2^, respectively. After 7 days of exposure, at low concentrations, the similar results to those for 3- and 5- day exposure were obtained, whereas, at the high concentrations (0.015, 0.025 and 0.05 mg AI/cm^2^), there were no significant differences in mortality levels between the tested surfaces. The complete mortality was achieved only on the concrete and laminate flooring surface at 0.015 and 0.025 mg AI/cm^2^, respectively, which the complete mortality was not reached on ceramic floor tile surface.

## 4. Discussion

Spinetoram has been reported to be effective against major stored product insects of stored-grain on different surfaces [62]. Our study is, however, the first to evaluate this insecticide against *A. obtectus* on different surfaces. The mortality of *A. obtectus* adults was significantly affected by spinetoram concentration, regardless of the surface type and exposure interval. Our overall data for spinetoram show that sufficient control can be achieved with increasing concentration after 5- and 7-day exposure, spinetoram treatments at 0.01 mgAI/cm^2^ and above concentrations achieved 100% or close mortality on concrete and laminate flooring surface. In the case of ceramic floor tile surface, spinetoram treatments at all tested concentrations did not result in 100% mortality. Our results indicated that 0.015 and 0.025 mg AI/cm^2^ concentration of spinetoram on concrete and laminate flooring surface, respectively, is enough to obtain the complete mortality of *A. obtectus* for 7-day exposure, whereas even the highest concentration on ceramic floor tile surface did not reach 100% mortality.

To our knowledge, this is the first study that examined the efficacy of spinetoram against *A. obtectus* on different surfaces. However, spinetoram has been evaluated against other stored product insect species with high adult mortality [56,62]. Vassilakos et al. [62], testing spinetoram at the three high concentrations (0.025, 0.05 and 0.1 mg AI/cm^2^) on concrete, ceramic tile, galvanized steel and plywood surface found that mortality levels of *T. confusum* did not significantly increase with increasing concentration. Similarly, Vassilakos and Athanassiou [57] reported that the increasing concentration from 0.025 to 0.1 mg AI/cm^2^ on concrete surface did not result in increased mortality of *T. confusum, S. oryzae* and *O. surinamensis* adults after 3- and 7-day exposure, except of *T. confusum* for 7-day exposure. These findings are not in agreement with our results for *A. obtectus* adults on concrete and ceramic tile surface. This may be due to the difference in tested insect species and formulation type. Previous studies reported that the insecticide formulation and the insect stage, as well as the insecticide exposure method are critical in insecticide toxicity [67,68,69]. Low mortality was recorded after 1-day exposure, on tested surfaces, whereas complete mortality was achieved after 5-day exposure. These results indicated that there was delayed mortality for *A. obtectus*. According to Vassilakos and Athanassiou [54], and Athanassiou et al. [70], spinosad and spinetoram are considered as relatively slow-acting insecticides in stored grain beetles. Vassilakos and Athanassiou [54], working with spinetoram-treated wheat reported that after 72 h of exposure, immediate mortality levels of *R. dominica* and *S. oryzae* were low.

Significant differences were noted among surfaces in mortality of *A. obtectus* adults for each exposure interval depending on spinetoram concentrations. After 3-, 5- and 7- day exposure, at the low concentrations (0.0025, 0.005 and 0.0075 mg AI/cm^2^), mortality levels on the ceramic floor tile surface were generally higher than those on the concrete and laminate flooring surface, whereas, at the high concentrations (0.015, 0.025 and 0.05 mg AI/cm^2^), those on the concrete and laminate flooring surface were significantly higher than on the ceramic floor tile surface. After 7-day exposure, the complete mortality was achieved only on the concrete and laminate flooring surface at 0.015 and 0.025 mg AI/cm^2^, respectively, whereas it was not reached on ceramic floor tile surface at any of spinetoram concentrations. In previous studies, Vassilakos et al. [62] noted that mortality of *T. confusum* adults on concrete and galvanized steel treated by spinetoram at two concentrations (0.025 and 0.05 mg AI/cm^2^) was higher than on ceramic tile and plywood in the absence of food, whereas there were no differences among surfaces for *S. oryzae* and *R. dominica*, due to the high efficacy of spinetoram in all surfaces, at relatively short exposures. Vassilakos and Athanassiou [56] also reported that there were no significant differences in the residual efficacy of spinetoram (0.025 and 0.1 mg AI/cm^2^) on concrete and galvanized steel surfaces against *S. oryzae* and *O. surinamensis* adults, whereas the higher mortalities of *T. confusum* adults were noted in the concrete surface than those in galvanized steel. Similarly, Toews et al. [41], evaluated spinosad at concentrations of 0.05 and 0.1 mg AI/cm^2^ in different surfaces, and found lower mortality levels of *T. confusum* and *T. castaneum* adults in treated unwaxed floor tile, steel and waxed floor tile surfaces than concrete surface. These findings are in agreement with our results for *A. obtectus* adults exposed to spinetoram on concrete, ceramic floor tile and laminate flooring surface.

Several studies indicated that the susceptibility of stored product insects to spinetoram-treated surfaces varied with insect species [62,71]. Vassilakos et al. [62] found that, after 5-day exposure, the complete mortality of *S. oryzae* was achieved on the concrete, ceramic tile and galvanized steel surface at 0.025, 0.05 and 0.1 mg AI/cm^2^, respectively, whereas that of *R. dominica* was obtained only on concrete surface at 0.025 mg/cm^2^ and none of spinetoram treatments on tested surfaces gave the complete mortality of *T. confusum*. Saglam et al. [59] reported that spinetoram at 0.05 and 0.1 mg AI/cm^2^ on concrete completely controlled *T. confusum* adults after 14-day exposure. The present study indicated that the 0.025 mg AI/cm^2^ concentration of spinetoram on concrete surface is enough to obtain the complete mortality of *A. obtectus* for 5-day exposure. These findings show that *A. obtectus* is apparently more susceptible to spinetoram-treated concrete than *T. confusum* whereas susceptibility of *A. obtectus* to spinetoram-treated is similar to that of *S. oryzae* and *R. dominica*.

Generally, the physical characteristics of the surfaces play an important role in residual efficacy of the insecticides. The insecticides applied in nonporous materials, such as steel and tile, are considered more effective than in porous materials, such as concrete and wood [61,63,72]. Porous surfaces, such as concrete, have low insecticide persistence than other nonporous surfaces [63,68,73,74,75,76]. Arthur [76] and Collins et al. [63] found that deltamethrin and organophosphates (OPs), respectively, were more effective on steel surfaces than on concrete. However, several studies, conversely, reported that some insecticides were more effective on concrete surface than on steel and ceramic tile surfaces. On the other hand, Arthur [77] showed that the pyrrole insecticide, chlorfenapyr, was more effective on concrete than on other nonporous surfaces, vinyl tile and plywood. Likewise, Vassilakos and Athanassiou [56] reported that spinetoram was more effective on concrete than on galvanized steel and ceramic tile. Similar results for *A. obtectus* adults were obtained in the present study. These findings suggest that different insecticide active ingredients show different efficacy and patterns of interactions with the surface on which they are applied, and that the distribution in different surfaces varies among insecticides [77].

In conclusion, our laboratory tests indicated that spinetoram at 0.025 and 0.05 mg AI/cm^2^ achieved satisfactory control at relatively short exposures by contact action of *A. obtectus* adults on three surfaces commonly encountered in legume storage facilities and warehouses. Thus, our results provide data on the insecticidal effect of spinetoram for the control of *A. obtectus* on various surfaces; however, its efficacy varies according to the surface type, exposure time and concentration. Even so, further research is required to evaluate more aspects on long-term protection of spinetoram against *A. obtectus* adults on different surfaces, doses and exposure times on the persistence of the residues and the behavior of this ingredient in “real world” conditions.

## Figures and Tables

**Figure 1 insects-13-00723-f001:**
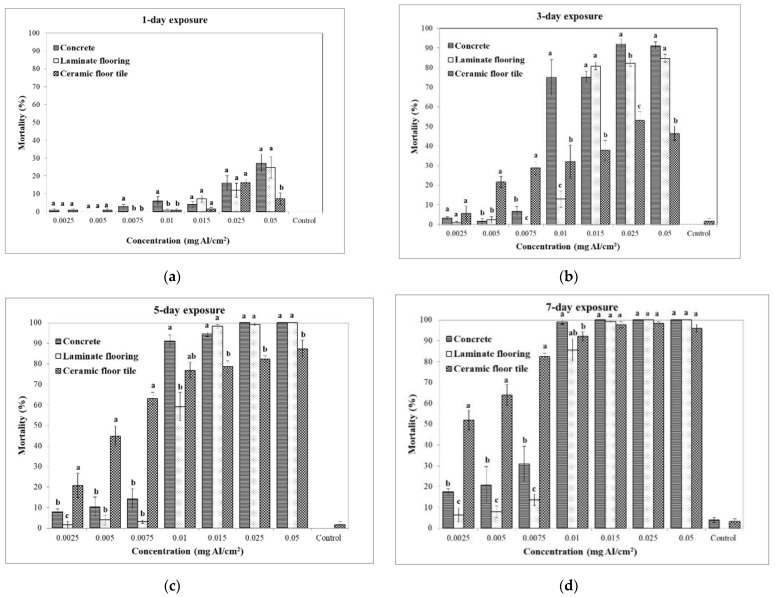
Mean corrected mortality (% ± SE) of *Acanthoscelides obtectus* adults after (**a**) 1-, (**b**) 3-, (**c**) 5- and (**d**) 7-day exposure to concrete and laminate flooring and ceramic floor tile surfaces treated with spinetoram at eight different concentrations. Means within each diagram and concentration are followed by the same lower-case letter are not differ significantly different (Tukey’s HSD test at 5% level).

**Table 1 insects-13-00723-t001:** ANOVA parameters for main effects and associated interactions for mortality of *Acanthoscelides obtectus* adults after four exposure intervals (total df = 104).

Exposure Interval	Source	df	F	P
1 day	Surface	2	7.02	0.002
	Concentration	6	33.88	<0.0001
	Surface x* Concentration	12	3.05	0.001
3 days	Surface	2	22.47	<0.0001
	Concentration	6	185.41	<0.0001
	Surface x Concentration	12	23.56	0.001
5 days	Surface	2	12.26	<0.0001
	Concentration	6	265.97	<0.0001
	Surface x Concentration	12	24.18	<0.0001
7 days	Surface	2	37.50	<0.0001
	Concentration	6	230.70	<0.0001
	Surface x Concentration	12	15.30	<0.0001

**Table 2 insects-13-00723-t002:** Mean corrected mortality (% ± SE) of *Acanthoscelides obtectus* adults after 1-, 3-, 5- and 7-day exposure to concrete surface treated with spinetoram at eight different concentrations.

Concentration (mg AI/cm^2^)	Exposure Interval (Day)			
	1 Day	3 Days	5 Days	7 Days
0	0 ± 0	0 ± 0	0 ± 0	4.0 ± 1.3
0.0025	0.8 ± 0.8 C *	3.2 ± 0.8 CD	8.0 ± 1.3 B	17.6 ± 1.3 B
0.005	0 ± 0 C	1.6 ± 1.6 D	10.4 ± 4.8 B	20.8 ± 8.9 B
0.0075	2.9 ± 1.2 C	6.7 ± 2.4 C	14.3 ± 4.9 B	30.8 ± 8.5 B
0.01	6.0 ± 2.5 CB	75.0 ± 9.1 B	91.0 ± 3.1 A	99.0 ± 1.0 A
0.015	4.0 ± 1.8 C	75.2 ± 2.9 B	94.4 ± 1.0 A	100 ± 0 A
0.025	16.0 ± 4.0 AB	92.0 ± 2.5 A	100 ± 0 A	100 ± 0 A
0.05	27.2 ± 4.9 A	91.2 ± 1.9 A	100 ± 0A	100 ± 0 A

* Means within each column with followed by the same upper-case letter are not differ significantly different (Tukey’s HSD test at 5% level).

**Table 3 insects-13-00723-t003:** Mean corrected mortality (% ± SE) of *Acanthoscelides obtectus* adults after 1-, 3-, 5- and 7-day exposure to laminate flooring surface treated with spinetoram at eight different concentrations.

Concentration (mg AI/cm^2^)	Exposure Interval (Day)			
	1 Day	3 Days	5 Days	7 Days
0	0 ± 0	0 ± 0	0 ± 0	0 ± 0
0.0025	0 ± 0 C *	0.8 ± 0.8 C	1.6 ± 1.6 C	6.4 ± 3.3 C
0.005	0 ± 0 C	2.4 ± 1.6 C	4.0 ± 2.5 C	8.0 ± 2.8 C
0.0075	0 ± 0 C	0.0 ± 0.0 C	3.2 ± 0.8 C	13.6 ± 2.7 C
0.01	0.8 ± 0.8 C	13.6 ± 4.1 B	59.2 ± 6.9 B	85.6 ± 5.2 B
0.015	7.2 ± 1.9 B	80.8 ± 1.9 A	98.4 ± 0.9 A	99.2 ± 0.8 A
0.025	12.0 ± 4.0 B	82.4 ± 1.6A	99.2 ± 0.8 A	100 ± 0 A
0.05	24.8 ± 5.9 A	84.8 ± 1.9 A	100 ± 0 A	100 ± 0 A

* Means within each column with followed by the same upper-case letter are not differ significantly different (Tukey’s HSD test at 5% level).

**Table 4 insects-13-00723-t004:** Mean corrected mortality (% ± SE) of *Acanthoscelides obtectus* adults after 1-, 3-, 5- and 7-day exposure to ceramic floor tile surface treated with spinetoram at eight different concentrations.

Concentration (mg AI/cm^2^)	Exposure Interval (Day)			
	1 Day	3 Days	5 Days	7 Days
0	0 ± 0	1.6 ± 1.6	1.6 ± 1.6	3.2 ± 1.5
0.0025	0.8 ± 0.8 C*	5.6 ± 3.7 E	20.8 ± 5.9 D	52.0 ± 4.6 D
0.005	0.8 ± 0.8 C	21.6 ± 2.9 D	44.8 ± 4.8 C	64.0 ± 4.9 CD
0.0075	0 ± 0 C	28.8 ± 3.4 CD	63.2 ± 2.9 BC	82.4 ± 1.6 BC
0.01	0.8 ± 0.8 C	32.0 ± 8.5 BCD	76.8 ± 3.9 AB	92.0 ± 2.2 AB
0.015	1.5 ± 0.9 C	37.9 ± 4.9 ABC	78.8 ± 2.7 AB	97.7 ± 1.5 A
0.025	16.1 ± 1.9 A	53.1 ± 4.4 A	82.3 ± 1.6 A	98.4 ± 0.9 A
0.05	7.2 ± 3.2 B	46.4 ± 3.5 AB	87.2 ± 4.3A	96.0 ± 1.8 A

* Means within each column with followed by the same upper-case letter are not differ significantly different (Tukey’s HSD test at 5% level).

## Data Availability

Data is contained within the article.

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
