# Peer review of "Efficacy of Spinetoram for the Control of Bean Weevil, Acanthoscelides obtectus (Say.) (Coleoptera: Chrysomelidae) on Different Surfaces"

_insects, 2022, doi:10.3390/insects13080723_

Round 1

Reviewer 1 Report

Overview:

The MS presents contact toxicity of seven doses of spinetoram on different surfaces against adults of Acanthocelides obtectus. The contribution is interesting and worthy of attention. It is well within the scope of the journal. The study concept is well designed and data is reliable. The manuscript is generally well written and structured.

Major points:

Discussion should be improved.

ln 274-287: This part of discussion is pretty similar with the results. Try to avoid repeating the results.

Minor points:

ln 13: Replace -four-  with three.

ln 30: -bean weevil; Acanthocelides obtectus- is the same. Increase visibility of MS by using different keywords.

ln 51: Delete words - protective and-

ln 57: Update the literature.

ln 64: Italicize -Saccharopolyspora spinosa-

ln 79-81: So far, author have used scientific and common names when mentioned species for the first time.

ln 133: Delite -,- after floor tile.

ln 324: Exclude underline.

Author Response

Dear Reviewer,

Many thanks for your interest on our work. We have revised it as your suggestion. Please find below our comments.

 Yours sincerely,

Özgür SaÄŸlam

Major points

Discussion should be improved.

ln 274-287: This part of discussion is pretty similar with the results. Try to avoid repeating the results.

REPLY: DONE.

Minor points:

ln 13: Replace -four-with three.

REPLY: DONE.

ln 30: -bean weevil; Acanthocelides obtectus- is the same. Increase visibility of MS by using different keywords.

REPLY: DONE.

ln 51: Delete words -protective and-

REPLY: DONE.

ln 57: Update the literature.

REPLY: DONE.

ln 64: Italicize -Saccharopolyspora spinosa-

REPLY: DONE.

79-81: So far, author have used scientific and common names when mentioned species for the first time.

REPLY: DONE.

ln 133: Delite -,- after floor tile.

REPLY: DONE.

ln 324: Exclude underline.

REPLY: DONE.

Reviewer 2 Report

The work by Saglam et al on A. obtectus exposure to doses of Spinetoram on different surfaces at different concentrations provides very useful information about using this insecticide formulation for this pest. The study was very well designed and the results are presented in a clear manner which can easily be understood.

One concern is that the English language needs to be more thoroughly proofread. The content of the paper is easily understandable, but there are pervasive examples of awkward grammatical constructions and missused words, which were too numerous to address in a review.

My only other minor comment would be that it would be interesting to describe whether or not there was any impression that their were sublethal effects of the insecticide observed when collecting the data.  The authors simply list that the insects were assessed for movement. I do not believe it is necessary to try to quantify any sublethal effects, but it would be useful if a casual description could be provided, which would help researchers who might want to follow up on this work in a way which considers any sublethal effects. An extra sentence or two in the paragraph spanning lines 151-159 would suffice.

Author Response

Dear Reviewer,

Many thanks for your interest on our work. We have revised it as your suggestion. Please find below our comments.

 Yours sincerely,

Özgür SaÄŸlam

My only other minor comment would be that it would be interesting to describe whether or not there was any impression that there were sublethal effects of the insecticide observed when collecting the data.

REPLY: Thank you for your comment. Yes, it is clear that our study was focused on the insecticidal effect of spinetoram and no experimentation was done for the sublethal effects (See lines 164-175).

The authors simply list that the insects were assessed for movement. I do not believe it is necessary to try to quantify any sublethal effects, but it would be useful if a casual description could be provided, which would help researchers who might want to follow up on this work in a way which considers any sublethal effects.

REPLY: DONE. See lines 163-166.

An extra sentence or two in the paragraph spanning lines 151-159 would suffice.

REPLY: DONE. The paragraph was modified. See lines 163-174.
